# Experimental Study on the Strength and Hydration Products of Cement Mortar with Hybrid Recycled Powders Based Industrial-Construction Residue Cement Stabilization of Crushed Aggregate

**DOI:** 10.3390/ma16124233

**Published:** 2023-06-07

**Authors:** Miaoyi Deng, Xiangbing Xie, Jingbo Zhuo, Yahui He, Kaiwei Wang

**Affiliations:** 1School of Civil Engineering and Architecture, Zhengzhou University of Aeronautics, Zhengzhou 450046, China; miaoyi_deng@163.com (M.D.); hyh1974647758@163.com (Y.H.); wangkaiweizua@163.com (K.W.); 2Yellow River Laboratory, Zhengzhou University, Zhengzhou 450001, China

**Keywords:** cement fly ash mortar, hybrid recycled powder (HRP), mix ratio dedign, mechanical properties, hydration product

## Abstract

The strength-formation mechanism for industrial-construction residue cement stabilization of crushed aggregate (IRCSCA) is not clear. To expand the application range for recycled micro-powders in road engineering, the dosages of eco-friendly hybrid recycled powders (HRPs) with different proportions of RBP and RCP affecting the strengths of cement-fly ash mortar at different ages, and the strength-formation mechanism, were studied with X-ray diffraction (XRD) and scanning electron microscopy (SEM). The results showed that the early strength of the mortar was 2.62 times higher than that of the reference specimen when a 3/2 mass ratio of brick powder and concrete powder was mixed to form the HRP and replace some of the cement. With increasing HRP content substituted for fly ash, the strength of the cement mortar first increased and then decreased. When the HRP content was 35%, the compressive strength of the mortar was 1.56 times higher than that of the reference specimen, and the flexural strength was 1.51 times higher; XRD and SEM studies of the hydrated cement mixed with HRP showed that the amount of CH in the cement paste was reduced by the pozzolanic reaction of HRP at later hydration ages, and it was very useful in improving the compactness of the mortar. The XRD spectrum of the cement paste made with HRP indicated that the CH crystal plane orientation index R, with a diffraction angle peak of approximately 34.0, was consistent with the cement slurry strength evolution law, and this research provides a reference for the application of HRP to produce IRCSCA.

## 1. Introduction

In cement production, the greenhouse gases produced account for 8% of global greenhouse gas emissions [1]. The academic literature has emphasized the significance of reducing dependence on Portland cement as a crucial sustainability challenge within the construction sector. Consequently, prior research has dedicated considerable effort to identifying alternative or supplementary cementitious materials that can partially or completely substitute cement in civil engineering applications [2,3]. As a result, extensive research has been conducted to assess the viability of various typical industrial by-products, such as fly ash, glass, marble, coal-bottom ash, and others as partial replacements for Portland cement [4,5]. The incorporation of recycled glass into concrete has been found to improve rheological properties, reducing the requirement for chemical admixtures, and increasing cost-effectiveness [6]. The addition of waste marble powder can significantly improve the bending performance of concrete beams, while also helping to reduce environmental pollution and resource waste [7]. Moreover, over the past decade, the utilization of alkaline activators alongside has emerged as a prominent area of research, aiming to entirely substitute Portland cement with supplementary cementitious materials (SCMs). This advancement has sparked significant interest in the field and has given rise to the progress of geopolymer concrete [6,8,9,10]. However, the feasibility of this approach in many parts of the world is questionable due to the different types of fly ash and SCM.

In a concrete mixture, cement, mineral admixtures, and water form the paste. The paste binds the sand and coarse aggregates into the whole through the coagulation and hardening of the cementitious material. Mosavi et al. conducted a comparative study to evaluate the performance of recycled concrete produced using the pressure-drop method and the mechanical method, based on the aforementioned theory. The findings of the research indicated that the utilization of the pressure-drop forming method effectively enhances the mechanical properties of recycled concrete [11]. Fly ash, a frequently utilized admixture in concrete, serves as the sixth component in the concrete mixture. Its purpose is to decrease the cement content and enhance the workability of the concrete [12,13]. Furthermore, fly ash finds extensive application in the field of road engineering. In China, the utilization of semi-rigid bases with inorganic binders in road construction offers significant advantages over flexible roads that rely on granular materials as road construction layers. Semi-rigid bases provide superior strength, load-bearing capacity, structural integrity, and rigidity, making them the preferred choice for more than 85% of road base engineering projects [14]. The production of fly ash as a by-product of coal-fired power stations presents notable economic and environmental challenges. By incorporating a certain amount of fly ash into CSCA, the hydration process can be effectively stimulated, enhancing interface adhesion and increased strength at later stages. This transformation marks a significant milestone in the conversion of industrial solid waste into valuable resources [14,15,16]. Moreover, the utilization of recycled brick-concrete composite micro-powder (RBCP) as a partial cement replacement in CSCA serves a dual purpose. It enables the efficient utilization of concrete and brick waste (C&BW) while aligning with the objectives of sustainable development strategies. To illustrate, let us consider a second-class highway with a road base thickness of 20 cm. In this case, a quantity of 1728 m^3^/km of recycled aggregate can be utilized for the production of cement-stabilized materials (CSCA) [17].

Furthermore, apart from the aforementioned upstream repercussions influenced by the choice of materials employed, the construction sector confronts mounting obstacles concerning the downstream consequences associated with the effective handling of the escalating volume of demolition waste engendered upon the culmination of a structure’s lifespan [18,19,20]. To illustrate, the annual production of construction and demolition waste (C&DW) in the United States and the European Union amounts to approximately 700 million tons and 800 million tons, respectively [21]. In the year 2020, the demolition of outdated structures in China resulted in the generation of an estimated 2.385 billion tons of solid waste, with concrete and clay bricks constituting approximately 50% of the overall composition. According to statistics, China has produced about 20 billion m^3^ of clay bricks in the past 50 years [22], and they will become the main component of construction waste in the next 50 years. The current construction waste or industrial solid waste generated during the demolition of buildings is mainly landfilled, stacked in open air, or made of recycled aggregates [20,23]. The treatment not only makes solid waste materials not fully utilized, but also causes environmental pollution. Moreover, many investigations have been made on the micro-properties [24], mechanical properties [25,26], and durability of recycled aggregate concrete [27,28]. The findings from the studies indicate that the utilization of recycled aggregate leads to a decrease in the properties of concrete waste. In addition, when recycled fine aggregate is added to the recycled concrete, the durability and mechanical properties are significantly reduced [29]. The publication of several review papers on the characteristics of recycled aggregate concrete has effectively expanded the range of applications for recycled aggregate [30,31]. Moreover, the crushing of construction and demolition waste (C&DW) enables the production of recycled aggregate, which can be subsequently utilized in road engineering, taking into account its mechanical and physical-chemical attributes. Nevertheless, during the process of recycling, a substantial quantity of waste powder (WP) is generated as a consequential by-product [28,32]. The efficient utilization of waste powder (WP) as a partial replacement for Portland cement offers a notable reduction in the necessity for landfilling and dumping, along with a decreased demand for the extraction of natural resources essential for cement manufacturing [19,33,34]. In short, based on the composite effect of material, the comprehensive utilization of construction waste and industrial solid waste materials has important engineering practical significance.

Research investigations have demonstrated that recycled brick powder (RBP), generated during the crushing of brick waste for the production of recycled aggregate, exhibits potential as auxiliary cementing material. In this context, the amorphous SiO_2_ and Al_2_O_3_ components within RBP can undergo reactions with the calcium hydroxide (CH) formed during the hydration process of cement [35,36]. The hydration by-product, encompassing silicate and aluminate compounds, exhibits an effective inhibitory effect against the detrimental consequences induced by the alkali–silicon reaction [37,38,39]. In a study conducted by Wild et al., various clay bricks sourced from European countries were collected and finely ground into micro-powder to serve as a partial substitute for Portland cement [40]. The chemical analysis conducted to evaluate the pozzolanic activity of the investigated brick types consistently demonstrated favorable results, indicating their significant pozzolanic reactivity. The findings regarding the strength development, derived from the examination of cement-mortar specimens, further corroborated the aforementioned outcomes [41]. In the investigations conducted by Mao [42] and Ma [43], an examination of the properties of recycled micro-powder revealed that its incorporation resulted in accelerated cement hydration reactions, improved compactness of the mortar’s microstructure, and consequently enhanced mechanical properties. It can act as a cement-hydration crystal nucleus, pozzolanic effect, and microaggregate filling effect, replacing part of fly ash or cement [44,45]. In addition, some researchers have studied the replacement of RBP for cement content in concrete. The results show that if the mix proportion of RBP in the concrete is designed correctly, the concrete strength will not be significantly affected. The outcomes of the study indicate that when the mix proportion of RBP in concrete is appropriately designed, there will be no substantial impact on the strength of the concrete. In the research conducted by Kim [46], the influence of RBP content on the compressive strength of mortar was investigated. The findings revealed that the strength of mortar exhibited the most significant increase during the later stages, with a remarkable 54.1% enhancement in compressive strength observed at 28 days compared to the 7-day strength. In a survey conducted by Kartini et al. [35], the utilization of RBP was implemented as a partial replacement for cement in concrete specimens. The experimental findings demonstrate that when the replacement amount is 10%, 20%, and 30%, the average strength is reduced by 4.4%, 8.4%, and 14.9%, respectively, compared with the reference sample without RBP. Bektas’s study showed that the replacement amount was as high as 25%, and the setting time would not be prolonged [47]. Furthermore, the strength of mortar is notably influenced by the particle size of the RBP. Zheng’s tests on mortar samples containing RBP graded in four different particle sizes (i.e., 0.04, 0.06, 0.1, and 0.3 mm) revealed that among the tested samples, the mortar with a particle size of 0.06 mm exhibited the highest strength [48].

On the other hand, the chemical composition of recycled concrete powder (RCP) or recycled brick powder (RBP) is mainly SiO_2_, Al_2_O_3_, and CaO [33]. Its early activity can be more remarkable than fly ash. Therefore, some scholars have studied the effect of two or more admixtures on the properties of cementitious materials. When maintaining a constant total content of hybrid recycled powder (HRP), substituting RBP with RCP results in an elevated activity index of the HRP. This observation suggests that RCP exhibits greater reactivity compared to RBP [33,49]. The commonly used HRP includes RCP and RBP. The experimental findings indicate that the early compressive strength of concrete, when incorporating HRP in varying proportions, surpasses that of concrete mixed solely with RBP or RCP [50]. The higher the proportion of the HRP replacing cement, the less cementitious will participate in the early hydration reaction, while the recycled powder does not play a substantial role in the earlier activity and has little contribution to the increase in early strength. As the curing age progresses, the hydration of cement becomes more thorough, resulting in an increased concentration of Ca(OH)_2_ within the concrete system. This, in turn, stimulates the reactivity of RCP and enhances secondary hydration. Consequently, this process effectively mitigates the decline in later compressive strength of the concrete [51]. Li [52] and others found that the combination of RBP, slag, and fly ash can effectively improve the 28 d flexural strength of mortar; Chen [53] and others believed that the mixture of RBP and fly ash could decrease the porosity of the composite cementitious material and increase enhanced erosion resistance.

Therefore, the pozzolanic characteristics of the RBP and RCP show that the HRP can be used as a cement supplement. Based on this, more and more interest has been attracted to exploring the use of the HRP. Furthermore, it is worth noting that fly ash exhibits potential applications as a concrete admixture and auxiliary cementing material in the context of cement stabilization of crushed aggregate (CSCA) for road engineering purposes. However, there are few reports on the interaction between solid waste materials and cement. However, there is a limited number of studies available that explore the impact of RBP, RCP, and fly ash on the strength of cement mortar as well as the underlying mechanisms of hydration. To respond to this need, in the presence of fly ash, it is of great theoretical significance to explore the comprehensive effect of RBP and RCP and its influence on the strength-formation mechanism of cement mortar, which is the basis for the evaluation of the corresponding mechanical properties and durability of the CSCA in road engineering, especially for the future development of HRP as concrete mineral admixture and auxiliary cementitious material.

## 2. Materials and Methods

### 2.1. Materials

The C&BW utilized in the production of RBP and RCP was sourced from the demolition of a factory district that had been in existence for 50 years. The factory district was situated in Er-qi town, located in Zhengzhou, Henan Province. This waste consisted of clay bricks, concrete, ceramics and so on. Ordinary Portland cement 42.5, in compliance with the Chinese standard JTG/T F20-2015 [51], was prepared for the experiments. Tap water, with an approximate temperature of 20 °C, was employed. River sand with a modulus of 2.9 was utilized as the sand component in the mortar samples.

The production of HRP involved a three-stage process subsequent to the initial processing of the waste clay bricks and concrete. During the initial stage, a crusher was employed to reduce the size of the waste bricks and concrete particles to achieve a fine particle size below 10 mm. Subsequently, the particles were sorted into various particle sizes through the utilization of a vibrating screen; these were categorized as recycled coarse aggregate. In the second stage, the material obtained from the first stage was introduced into a ball-grinding mill featuring a circular cavity. This process aimed to produce fine aggregates with a maximum size of 2.36 mm, thereby classifying them as recycled fine aggregate. Subsequently, the output obtained from the second stage was subjected to an electromagnetic sample pulverizer to produce fine powders with maximum sizes of 45 μm, which were classified as HRP. The preparation process for HRP through comprehensive processing of the C&BW is depicted in Figure 1. The overall processing time, which encompassed crushing, sieving, and grinding, amounted to approximately 6–8 min.

### 2.2. Experimental Design

In order to examine the progression patterns of these recycled materials and comprehend the mechanical and hydration mechanisms of cement-fly ash mortars incorporating HRP, the proportions of RBP and RCP were chosen as variables, while maintaining a constant water/binder (w/b) ratio of 0.45 and a constant binder/sand (b/s) ratio of 0.3. The composition of RBP and RCP exhibits similarities to that of fly ash, characterized by elevated levels of SiO_2_ and Al_2_O_3_, which contribute to the enhanced pozzolanic activity. Consequently, it is anticipated that appropriately sized RBP and RCP particles can serve as SCM in concrete. Upon extensive grinding, both RBP and RCP exhibit a notably heightened reactivity. Consequently, these materials are harnessed as SCM during the process of concrete formulation. Moreover, scholars worldwide have conducted extensive research on the micro-performance, mechanical characteristics, and long-term durability of concrete incorporating RBP and RC. Under the same alternative mass fraction, as the curing period progresses, the strength of recycled composite micronized cement sand gradually increased in the presence of fly ash, and when the regenerated composite powder completely replaced fly ash, the strength of the mortar with HRP increased at 3 d and 7 d, and the strength decreased by 28 d, which indicated that the regenerated composite micro-powder mainly played a microaggregate filling effect in the early strength formation of the specimen, and with the gradual extension of the age, the later strength decrease was not significant because the active ingredient could react with cement hydration products. The plastic properties of mortar with HRP are improved.

Based on the results of Xiao [19], Tang [33], Zhu [39], Li [49], Kanellopoulos [54], Sun [55], Ma [56], and others, 5%, 15% and 25% of the cement was replaced by HRP, and 5%, 15%, 35%, and 100% of the fly ash was replaced by HRP. A unique identification code was assigned to each specimen, such as JZ25B3C2, which denoted a 25% replacement of cement and a mass mixing ratio of RBP/RCP at 3/2. Furthermore, the notation HRP-I was employed to indicate a mass ratio of 3/2 for RBP and RCP. The mass ratio of cement and fly ash was 3.5%:5% in the CSCA reference specimen, which was denoted as JZ (Table 1).

### 2.3. Preparation of Mortar and Paste Samples

The mortars were cast into prismatic molds measuring 40 mm × 40 mm × 160 mm for casting and demolded after 24 h. The mortar specimen color changed with the changes in the RBP replacement rate, as shown in Figure 2; with increased RBP replacement rates, the color of the cement-mortar specimens gradually changed from grey to pink (from right to left). All of the samples were placed in a testing environment room until they reached a certain age (3 d, 7 d, or 28 d), and they were held at a constant temperature of approximately 20 °C ± 1 °C and under a constant relative humidity (RH) ≥ 90%.

To analyze the effect of the HRP on the strengths of cement-fly ash mortars with cement paste (CP) as the benchmark, cement-fly ash paste (CFP), cement-fly ash recycled brick powder paste (CFBP), cement-fly ash recycled concrete paste (CFCP), and cement-fly ash recycled brick powder concrete paste (CFBCP) were used as research objects. The paste specimens were prepared with an NJ-160B mixer and then modelled in rectangular molds measuring 40 mm × 40 mm × 160 mm. The curing conditions employed for the paste were analogous to those employed for the mortar specimens. The relative compressive strength of the composite micronized concrete was evaluated at 3, 7, and 28 days, considering various substitution rates.

### 2.4. Sample Testing

The particle size distributions of both RBP and RCP were determined using a laser diffraction particle size analyzer. The chemical compositions and mineralogical compositions of both RBP and RCP were analyzed using X-ray fluorescence spectrometry (XRF) and X-ray diffraction (XRD) techniques. The flexural strength and unconfined compressive strength of the investigated cement-mortar samples were determined in accordance with the specifications outlined in GB/T17671-2021. The respective strengths were measured to evaluate the mechanical performance of the samples. The specimens were subjected to flexural strength and uniaxial compressive loads, with loading rates of 50 N/s and 2500 N/s, respectively, until failure occurred. Scanning electron microscopy (SEM) was used to study the micromorphology of the HRP, and the cement-fly ash mortars contained HRP. The SEM sample preparation and observation procedures were conducted as described below: After conducting the 28-day compressive strength test on the cement-fly ash mortar samples containing HRP and the cement-fly ash mortar samples without HRP, three cubes measuring 1 cm × 1 cm × 1 cm were carefully extracted from the central region of each specimen. The extracted cubes were immersed in anhydrous ethanol to halt the hydration process, followed by drying in an electric heating blast-drying oven maintained at 60 °C. Once cooled, the samples underwent gold coating before being subjected to SEM imaging. The mineralogical compositions of the respective pastes were determined with XRD analysis, employing a scan rate of 0.02°/min.

## 3. Results and Discussion

### 3.1. Particle Size Distributions

The particle size distributions of RBP and RCP reflect their gradations. The laser particle size analyses of the cement, recycled coarse aggregate (RCP), recycled fine aggregate (RBP), and fly ash are depicted in Figure 3b. The particle sizes of these powders ranged from 0.1 μm to 100 μm. The leftward shift of the distribution peak indicated an improvement in the gradation of RP. This observation suggested that the particle size distribution of recycled fine aggregate RBP was superior to that of recycled coarse aggregate RCP. The particles of recycled brick powder are finer and evenly distributed, the particles of recycled concrete powder are coarser, the particle size of the former is smaller than the latter, and the particle size distribution of RBP is better than that of RCP. Figure 3b shows that the RBP particles were fine and evenly distributed. In contrast, the RCP particles were coarse. The d50, d32, and d43 values for RBP were 12.639 μm, 3.900 μm, and 20.600 μm, respectively. The RCP values were 19.053 μm, 4.584 μm, and 23.828 μm, respectively. The average particle size of the cement was 18 μm [19,54], and the average particle size of the RBP was between those for the RCP and the cement. In RBP and RCP, the contents of particles smaller than 10 μm were 41.49% and 32.52%, respectively, and the contents of particles smaller than 45 μm reached 81.15% and 80.30%, respectively. According to Chinese standard GB/T 1596-2017 [57], both RBP and RCP met the particle size requirements for secondary fly ash.

### 3.2. Chemical and Mineral Compositions

Table 2 reveals that the SiO_2_ content of the RBP closely resembled that of the fly ash. Additionally, the CaO, Al_2_O_3_, and Fe_2_O_3_ contents of RBP were found to lie between those of the cement and fly ash. This observation suggests that RBP exhibited a favorable distribution of oxides. Based on the diffraction peaks shown in Figure 4, RBP contained SiO_2_ crystals, which can be combined with the Ca(OH)_2_ generated by cement hydration to generate C-S-H and C-A-S-H, calcium sulfoaluminate hydrate [39,58,59]. Li et al. also indicated that fine SiO_2_ increased the compactness and strength of a mortar sample. Table 2 shows that the RPC contained CaO, which reacts with water to form the alkali activator Ca(OH)_2_, and Ca(OH)_2_ improves the activity of the RBP. Moreover, both the RBP and RCP contained SiO_2_ and Al_2_O_3_. Hence, both powdered materials exhibited pozzolanic activity, which effectively improved the late strengths of the mortars. In addition, there was a small amount of albite in the recycled concrete powder, which may have arisen from mixing of fine aggregate stone chips; additionally, the recycled brick powder contained a large amount of albite, and albite was one of the main components of the clay bricks. The presence of CaCO_3_ in RCP has the potential to reduce the induction period of C_3_S and actively participate in the hydration reaction, leading to the formation of carboaluminates. This, in turn, contributes to the refinement of pore structures in cement pastes [60]. Nonetheless, the XRD pattern of the RCP did not exhibit diffraction peaks corresponding to hydrated calcium silicate or hydrated calcium aluminate. This observation suggests that the cement particles present in RCP had undergone hydration. In summary, RBP and RCP showed specific activities and replaced some of the cementitious materials or mineral admixtures.

### 3.3. Micro-Structure Analysis

The hydration behavior of cementitious materials is intricately linked to their macroscopic and microscopic characteristics. Good hydration leads to higher concrete strength. Enhanced fineness of the recycled powders led to an increase in the concentration of amorphous oxides within the cementitious materials, thus improving their hydration reactivities [49,54,61] Finer RBP and RCP exhibited higher activities, and when the RBP content of the HRP exceeded 75%, the strength activity index was more than 70%. Therefore, in this paper, the maximum possible amount of RBP was used, the HRP was prepared with mass ratios (brick/concrete, B/C) of 5/0, 4/1, 3/2, and 0/5, and microstructural observations were conducted with SEM. Figure 5 shows SEM images of fly ash and the types of HRP.

Figure 5 presents the microstructures resulting from different RBP and RCP mass ratios, and the samples were relabeled as a, b, c, and d. In comparison to the microstructure of fly ash, the microstructure of the HRP particles was exceptionally irregular; most were block-shaped and had edges and corners, and there were many microcracks on the surfaces. This enhanced the workability of the concrete mixture with HRP, which is normally lower than that of concrete mixtures without HRP [33,38]. As shown in Figure 5a,b, both RBP and RCP were compounded with different mass ratios. The surface of the HRP with a mass mixing ratio of 4/1 was denser, while the sample made with a mixing ratio of 3/2 was loosely bound and had a rough surface; the rough surface and more porous structure made it easier for the cement paste to penetrate the HRP. Distinct dissimilarities were evident in the SEM images of RCP and RBP, as illustrated in Figure 5c,d, respectively.

### 3.4. Mechanical Properties Analysis

In the presence of fly ash, RBP (B) and RCP (C) were compounded with mass ratios of 3/2 and 4/1 to replace some of the cement cementitious materials or fly ash, and the samples were denoted HRP-I and HRP-II.

#### 3.4.1. Flexural Strength

The effects of the RBP and RCP contents on the flexural strengths of the cement-fly ash mortar at different ages are shown in Figure 6.

Figure 6 shows the 3 d, 7 d, and 28 d flexural strengths of the cement-fly ash mortar with various RBP replacement proportions and different RCP replacement proportions. With increasing contents of the recycled micro-powder replacing the cement cementitious material and fly ash, the flexural strengths generally first increased and then decreased. Figure 6a,b show that compared with the early strength of the JZ sample when the ratio of recycled micro-powder replacement was below 15%, the flexural strength of the mortar increased after adding RBP and RCP. Furthermore, the increment in early strength observed in the mortar incorporating RBP exceeded that of the mortar utilizing RCP. This phenomenon can be attributed to the smaller average particle size of the RBP compared to that of the cementitious powder. RBP formed microaggregates to improve the particle gradation of the cementitious materials, thereby improving the compactness of the mortar [53]. At a replacement ratio of 15%, the mortar exhibited the highest flexural strength value. Figure 6e shows that the flexural strength of the HRP-I replacement fly ash was consistent with the evolution trend seen for the mortar containing RBP and RCP. Nevertheless, at a mass fraction of 35%, the 28-day flexural strength of the cement-fly ash mortar reached 5.90 MPa, indicating a remarkable increase of 51.28% compared to the initial strength. This resulted in a flexural strength that was 1.51 times higher than that of the JZ specimen. The findings clearly demonstrated that the HRP-cement-fly ash system exhibited the highest mechanical strength, leading to a significant improvement in the overall strength of the mortar.

Figure 6c,d illustrate the associations between the flexural strength and the replacement ratios of HRP. The strength of the cement-fly ash mortar containing HRP exhibited a gradual increment with the progression of the curing age. Furthermore, the strength exhibited by the mortar specimens containing HRP-I surpassed that of the mortar specimens containing HRP-II. Generally, with increasing amounts of cement binder replaced by the HRP, the strength of the mortar specimen first increases and then decreases. The optimal dosage of HRP positively influenced the flexural strength of the mortar. The suitable replacement ratio for both HRP-I and HRP-II was determined to be 15%. The HRP-I content was appropriate because HRP-I has high pozzolanic reactivity and microaggregate filling effect, which is due to a reaction of the SiO_2_ and Al_2_O_3_ in the recycled micro-powder with the Ca(OH)_2_ generated by hydration of the cement clinker to produce the pozzolanic effect and generate hydraulic C-S-H and C-A-S-H [38,45,54,58]. The fine particles in the recycled micro-powder were uniformly distributed in the cement slurry to form the microaggregates and improve the compactness of the mortar [54,58]. These two factors compensate for the notable decline in compressive strength resulting from the reduced proportions of cementitious materials.

#### 3.4.2. Compressive Strength

The paramount mechanical property parameter, compression strength, assumes a position of utmost significance. In the context of hardened mortar, this parameter assumes a pivotal role primarily attributed to its capacity for bonding and facilitating load transfer. The outcomes depicted in Figure 7 provide evidence that the compressive strength exhibits a characteristic pattern of initial augmentation followed by subsequent diminishment in response to increasing replacement ratios of RBP, RCP, and HRP. Nonetheless, it is worth noting that when the replacement ratio of RBP, RCP, or HRP reaches either 25% or 100%, the compressive strength of the mortar incorporating these replacements falls significantly below that of the reference samples, specifically the JZ samples. This phenomenon arises due to the inclusion of RBP, RCP, or HRP, which leads to a reduction in the quantity of hydration products. Consequently, the integrity of cementitious materials experiences a decline upon incorporating RBP or RCP. Furthermore, it is noteworthy that the adverse impact exerted by RBP, RCP, or HRP on the compressive strength diminishes as the duration of the curing process is extended. This can be attributed to the enhanced progression of the pozzolanic reaction within the RBP, RCP, or HRP mortar, facilitated by the prolonged curing time. Additionally, it should be noted that elevating the HRP-I replacement ratio proves to be an effective strategy for enhancing the compressive strength, particularly when the replacement ratio remains below 35%.

The 3 d compressive strength of the mortar exhibited the most pronounced increase, as evident from the findings presented in Figure 7a,b. Those of the mortars with RBP or RCP increased by 19.08% and 14.29%, respectively. The 28 d compressive strength of the RBP mortar was recorded at 20.9 MPa, indicating a decrease of 2.39% in comparison to the 21.40 MPa achieved by the JZ specimen. Conversely, the RCP mortar exhibited a compressive strength of 22.0 MPa, signifying a 2.80% improvement over the compressive strength of the JZ specimen. Furthermore, it is noteworthy that when the HRP replacement ratio remained below 15%, the compressive strength of the HRP-incorporated mortar surpassed that of the JZ specimen. This occurred because the HRP has a physical filling effect in the early stages. With longer curing ages, the pozzolanic effect of HRP gradually increases. Among them, RCP and RBP in the HRP provided the crystal embryo effect [19] and hydration nucleation effect [46] for the cement hydration process, respectively. The composite effect of the two was most effective with a mass ratio of 3/2. When the RBP and RCP were mixed, the compressive strength was greater than those seen with RBP or RCP alone, which once again proved that the RBP and RCP composite showed increased activity. CaO in the RCP reacted with water to generate the alkali activator Ca(OH)_2_, which then reacted with SiO_2_ and Al_2_O_3_ in the RBP to generate hydraulic C-S-H3745 and C-A-H [54,58]; this also showed that the RCP was most effective in activating the RBP at this mass ratio.

Compared with the JZ specimen, when the HRP-I replacement ratio was 15%, the compressive strength of the mortar at 3 d and 7 d after casting were measured to be 7.64 MPa and 16.8 MPa, respectively, and the 28 d compressive strength was 23.90 MPa, a maximum increase of 11.68%. These results indicated that HRP-I prepared with a mass ratio of 3/2 effectively improved the early strength, which was consistent with the microstructure of the HRP shown in Figure 5. Furthermore, RBP and RCP exhibited pozzolanic activity. The CaO present in RCP underwent a reaction with water, resulting in the formation of the alkali activator Ca(OH)_2_. This activation process further facilitated the pozzolanic effect of RBP, consequently enhancing the strength of the cement-fly ash mortar.

Figure 7e shows the relationship between the compressive strength and HRP-I replacement of the fly ash. When the mass fraction was 35%, the 28 d compressive strength of the cement-fly ash mortar was 33.50 MPa, so the compressive strength had increased by 56.54% and was 1.56 times higher than that of the JZ specimen. The results showed that the mechanical strength of the HRP-cement-fly ash system was the best, and the strength of the mortar was improved. However, when HRP completely replaced the fly ash, the 3 d and 7 d strengths of the mortar specimens increased, and the 28 d strength decreased by 4.39%, indicating that HRP mainly played the role of microaggregate filling in the early strength. With increasing age, the strength of the cement mortar decreased slightly due to reactions of the active ingredients with the cement hydration products, and the plastic properties of the cement mortar were improved. These ratios were appropriate because the microaggregate effect, filling effect, and pozzolanic effect of the three materials were utilized. The compressive strength of the reclaimed HRP-cement-fly ash mortar met the technical requirements for inorganic binder-stabilized base courses used in road engineering [62].

### 3.5. Further Microstructural Analysis

#### 3.5.1. Effect of the HRP-I on the Microstructure of Cement-Fly Ash Mortar

Figure 8a shows that white flocculent substances were attached to the surfaces of the fly ash spherical particles in the cement-fly ash mortar. This material is a hydraulic cementitious material, such as C-S-H and C-A-H, and it is formed by reactions between SiO_2_ and Al_2_O_3_ in the fly ash with the cement hydration product Ca(OH)_2_, which is consistent with the conclusions of Du [63], Hardjito [8], and Narmluk [64]. Needle-like substances appeared between the spherical particles of the fly ash, and these comprised ettringite (AFt) formed by the reaction of Al_2_O_3_ and Ca(OH)_2_ in fly ash. A comparison of Figure 8a,b showed that there were holes in the internal micromorphologies of the cement-fly ash mortar made with HRP and the cement-fly ash mortar made without HRP. However, the cement-fly ash mortar containing HRP was forming a relatively stable dense structure, which showed that HRP filled the microaggregates. In addition, the CaO in the RCP reacted with water to generate the alkali activator Ca(OH)_2_, which promoted a secondary hydration reaction between the active component of the RBP and the cement hydration product. The combined effects of RBP and RCP increased the hydration products formed in the mortar with HRP, and the thicknesses of white flocculated substances attached around the fly ash particles in Figure 8b was larger than for those in Figure 8a. Moreover, the presence of numerous fibrous crystals was observed to be distributed among the hydration products, such as C-S-H gel, as depicted in Figure 8b. These fibrous crystals effectively enhanced the densification and compactness of the cement paste. The cumulative effects of HRP-I resulted in the enhanced strength of the cement-fly ash mortar. Figure 8c displays the elemental map of the cement-fly ash mortar and the cement-fly ash mortar with HRP. It is evident that the calcium content decreases while the silicon content increases with an increase in the HRP replacement ratio. This trend can be attributed to the lower calcium and higher silicon content present in HRP compared to cement.

#### 3.5.2. Effects of HRP-I on the Hydration Products of Cement Paste

Figure 9 shows that the main hydration products formed at 28 d in the different cement pastes were calcite (CaCO_3_), ettringite (AFt), calcium hydroxide (CH), SiO_2_, C-S-H, and C-A-H. SiO_2_ came from the fly ash, CaCO_3_ came from the HRP, and calcium hydroxide (CH) was derived from the cement hydration products. The hydration process was similar to that described in the literature [54,60] and mainly Aft, C-S-H gel, and C-A-H gel were formed, which strengthened the cement-fly ash mortar with HRP. Compared with the CH diffraction peak intensity for the CP, the CH diffraction peak for CFBP had the lowest intensity and the largest width, indicating that CFBP had the lowest pozzolanic activity. Compared with CFP, the diffraction peak intensities for SiO_2_ and Al_2_O_3_ in the CFBCP were reduced because of the reactions occurring between the SiO_2_ and Al_2_O_3_ in the HRP with the cement hydration product CH to form the C-S-H gel and C-A-H gel. Figure 8 shows that CH had formed layered or flaky hexagonal crystals, and the arrangement affected the compressive strength of the cement mortar [33,34,60]. Therefore, the characteristic diffraction peak intensity was used to calculate the orientation index *R* for the calcium hydroxide (CH). When the *R* value was larger, the orientation of the CH was more substantial, and the compressive strength was lower. In this paper, the CH crystal plane with a diffraction angle of 18.1° (001) was selected as the reference plane, and the crystal plane orientation index *R* with a diffraction angle of 34.0° (101) was:(1)$$R=\frac{Intensity\left(001\right)}{Intensity\left(101\right)\times 0.74}.$$

Table 3 shows that the R value of the cement paste was the smallest, and the R values of all cement paste samples containing fly ash were greater than 1.622. According to the laboratory test results, the maximum compressive strength of the cement paste after 28 days was 15.12 MPa, which was greater than the strengths of the cement-fly ash paste and the cement-fly ash paste with HRP. The corresponding compressive strength decreased after using fly ash or HRP to replace some of the cement. To determine the relationship between R and the compressive strength, a linear plot and an exponential fitting function were used to perform a regression analysis on the data, and the results are shown in Figure 10. Figure 10 shows that the correlation coefficient R^2^ was 0.979 when an exponential function was used to fit a plot of the orientation index R versus the compressive strength of the mortar. With a linear function, the correlation coefficient R^2^ was 0.925. This observation signifies that the compressive strength of the paste is influenced by a multitude of factors. The orientation index R served as an indicator to gauge the impact of HRP on the compressive strength within the cement-fly ash system. In addition, in the presence of fly ash, the R values of the HRP pastes were lower than the R values for the RBP or the RCP, and the R value for RBP was greater than that for RCP. This occurred because the SiO_2_ in the RBP and fly ash accelerated the hydration processes in the early stages of cement hydration, which resulted in decreased intensities for some of the diffraction peaks. This was consistent with the change law for the CH diffraction peak eigenvalues at diffraction angles of 18.1° (001) and 34.0° (101). Therefore, the orientation index R indicated the effect of the HRP on the compressive strength of the cement-fly ash system. The orientation index R of CH can be used to reveal the strength evolution law for the HRP-cement-fly ash system.

## 4. Conclusions

The utilization of construction solid waste is an effective aspect of green and sustainable economic development. The comprehensive experimental study described herein indicates the viability of green recycling and using HRP to replace Portland cement and fly ash. The mechanical and hydration mechanisms of the cement-fly ash mortar were investigated by varying the proportions of RBP and RCP in HRP, and the replacement ratio of HRP. Based on the outcomes derived from this study, the subsequent conclusions can be drawn.

(1)HRP has specific activities and can be used as auxiliary cementitious material to replace some cement. In the presence of fly ash with increasing HRP content, the strength of the cement mortar first increased and then decreased, and the early strengths of the cement-mortar specimens with single RBP were better than those of the cement-mortar specimens with RCP. When the single content was lower than 15%, the maximum increase in 3 d compressive strength was 19.08%, and the maximum decrease in 28 d compressive strength was 2.39%.(2)HRP can be used as a mineral admixture to replace some of the fly ash. They were mixed with a mass ratio of 3/2 to form the HRP, which effectively improved the strength of the mortar specimen. When the RBP, RCP, and fly ash were mixed with the mass ratio 21/14/65, the flexural strength of the mortar was 1.51 times higher than that of the cement-fly ash specimen, and the compressive strength was 1.56 times higher at 28 d.(3)In the presence of fly ash, the HRP mainly exerted the microaggregate effect and promoted the cement hydration reaction, which significantly improved the microstructural compactness of the mortar and the thickness of the C-S-H on the surfaces of the fly ash particles.(4)The pozzolanic effect operating on the HRP in the later stages reduced the intensities of the CH diffraction peaks. The crystal plane orientation index R with a diffraction angle of 34.0° (101) was consistent with the evolution law for compressive strength.

## Figures and Tables

**Figure 1 materials-16-04233-f001:**
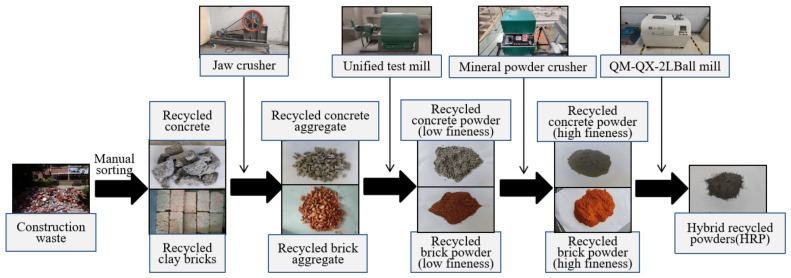
Preparation of HRP.

**Figure 2 materials-16-04233-f002:**
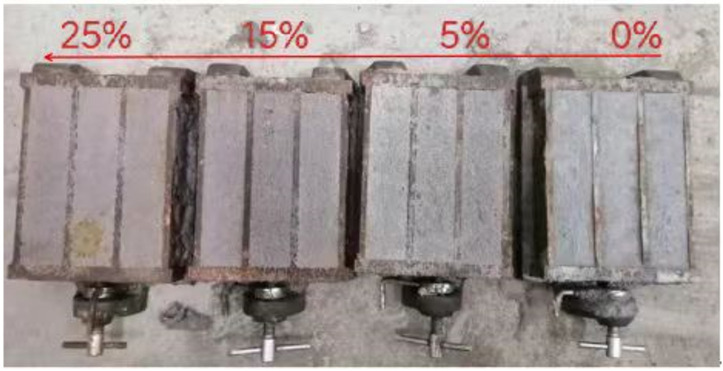
Evolution of the color of the mortar samples with the substitution rate of RBP.

**Figure 3 materials-16-04233-f003:**
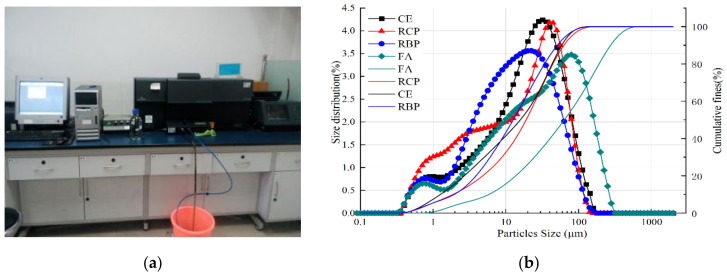
Particle size distribution. (**a**) LS-230 laser particle analyzer; (**b**) Particles size distribution of materials.

**Figure 4 materials-16-04233-f004:**
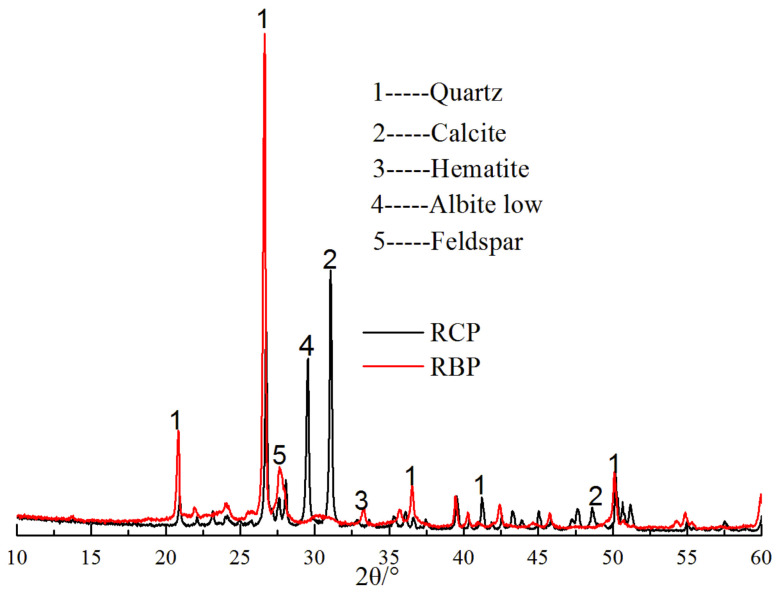
The XRD patterns of RBP and RCP.

**Figure 5 materials-16-04233-f005:**
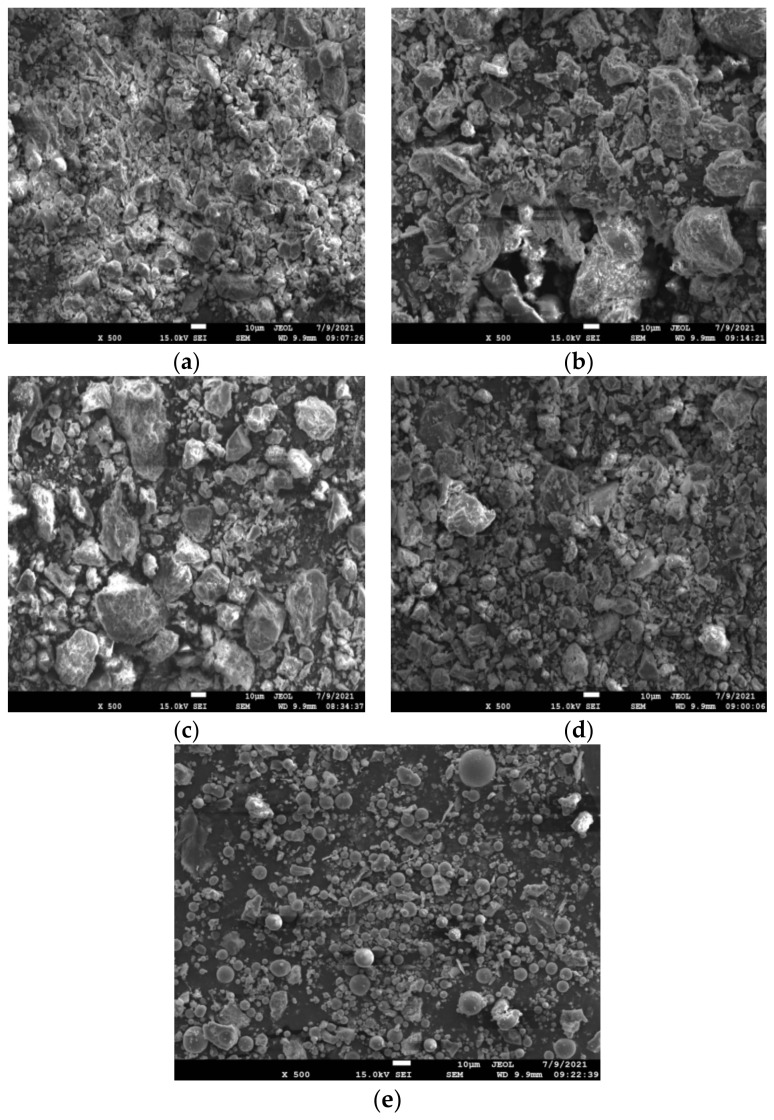
Microstructure of the HRP and fly ash. (**a**) HRP-II; (**b**) HRP-I; (**c**) RBP; (**d**) R€; (**e**) Fly ash.

**Figure 6 materials-16-04233-f006:**
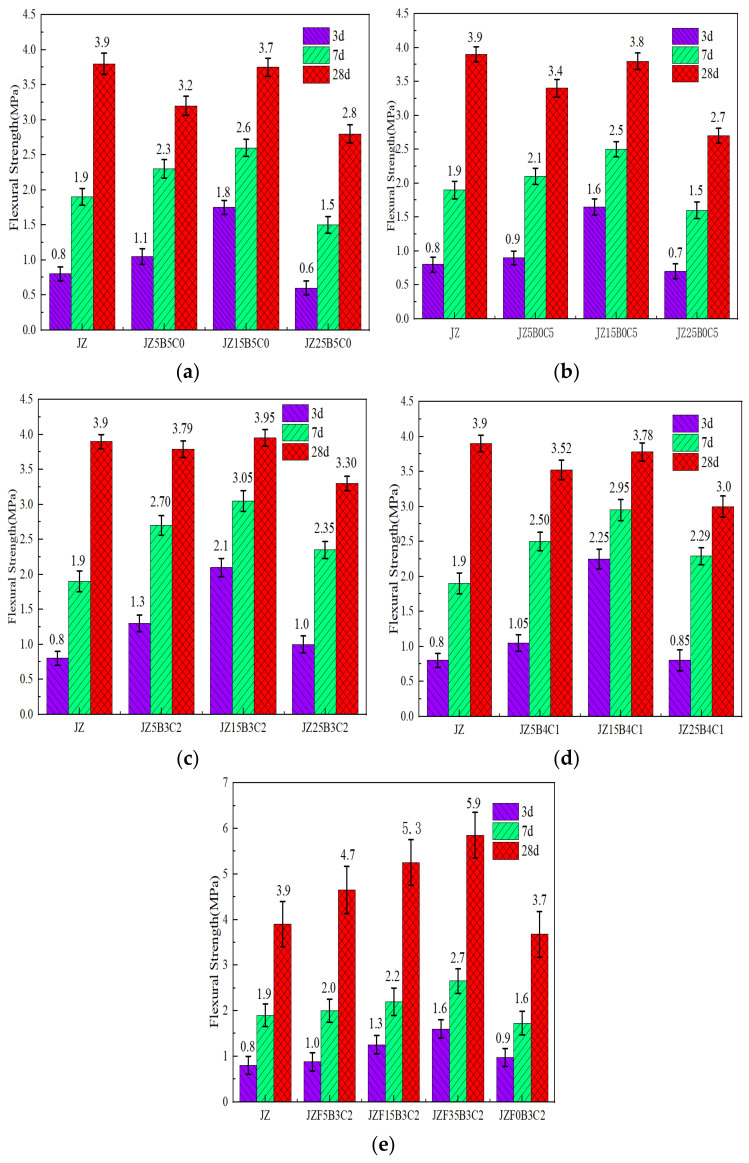
Flexural strength of cement-fly ash mortar containing the RBP or RCP. (**a**) mortar with RBP; (**b**) mortar with RCP; (**c**) mortar with HRP-I; (**d**) mortar with HRP-II; (**e**) HRP-Ireplacement fly-ash.

**Figure 7 materials-16-04233-f007:**
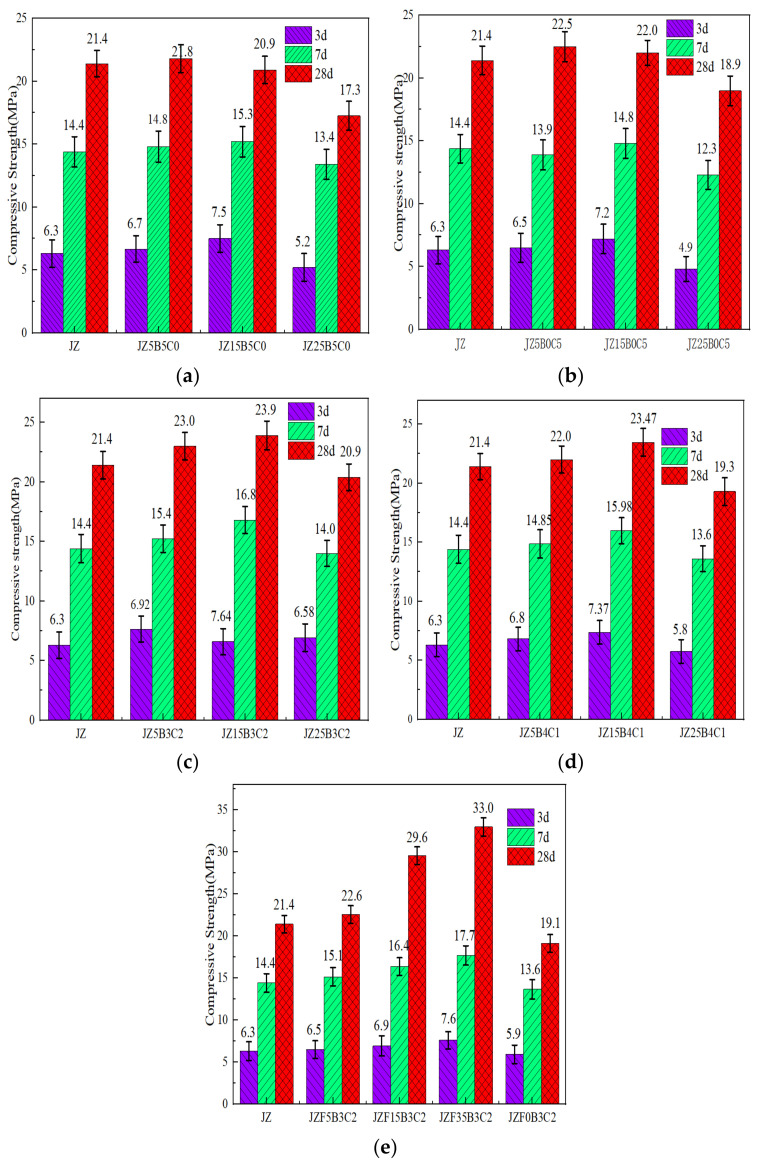
Compressive strength of cement-fly ash mortar containing the RBP or RCP. (**a**) mortar with RBP; (**b**) mortar with RCP; (**c**) mortar with HRP-I; (**d**) mortar with HRP-II; (**e**) HRP-I-replacement fly ash.

**Figure 8 materials-16-04233-f008:**
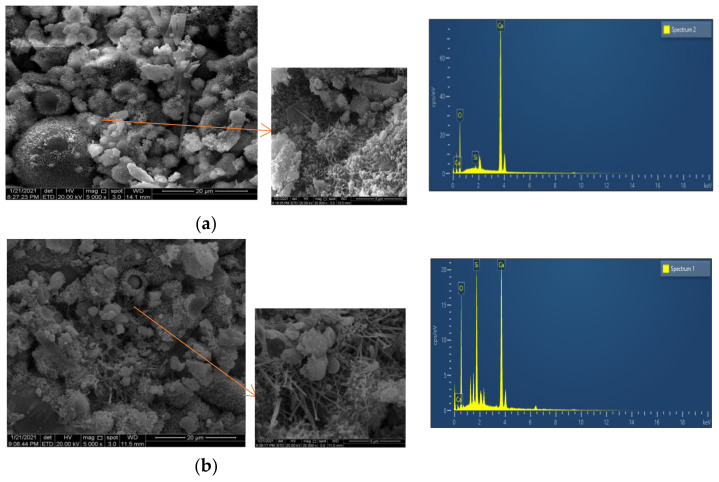
Micro-morphology of cement-fly ash mortar with HRP-I (28 d). (**a**) Cement-fly ash mortar specimen; (**b**) JZF35 cement-mortar specimen.

**Figure 9 materials-16-04233-f009:**
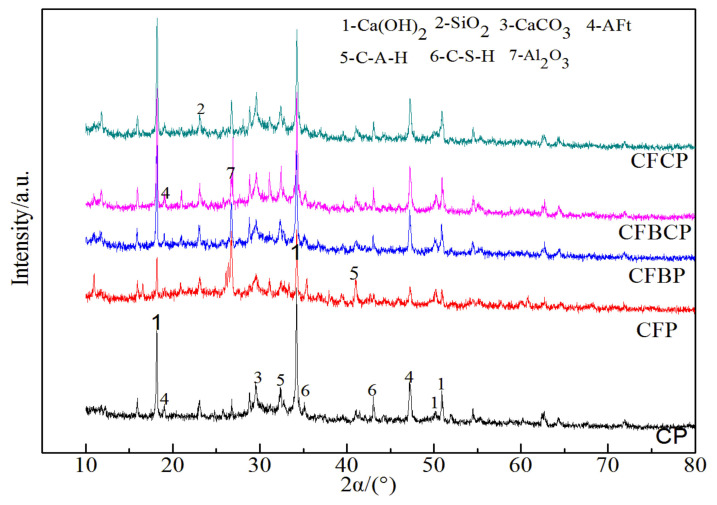
XRD patterns of the types of cement slurry with the fly ash.

**Figure 10 materials-16-04233-f010:**
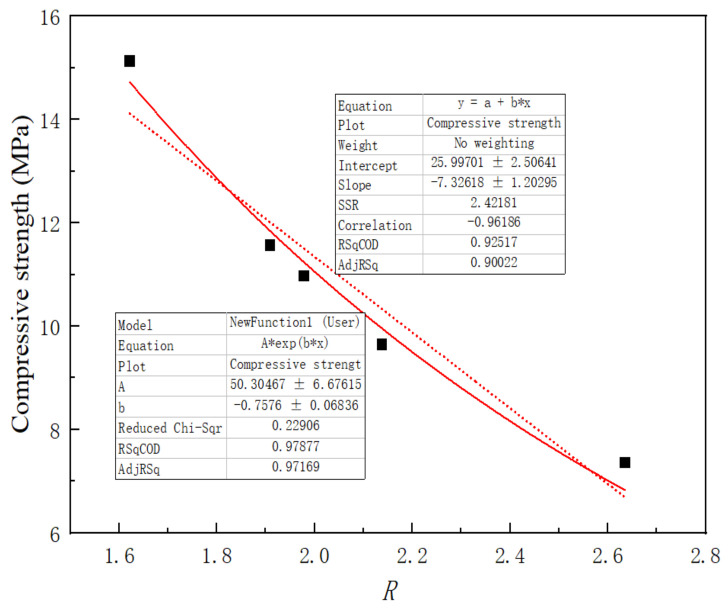
Variation trend of compressive strength of the R value.

**Table 1 materials-16-04233-t001:** Mix proportion of specimens.

Type	Replacement Rate	Cement	HRP (RBP:RCP)/(kg/m^3^)						Fly Ash
			5:0	4:1 (HRP-II)		3:2 (HRP-I)		0:5	
JZ	0	185.30	-	-	-	-	-	-	264.70
JZ5B_x_C_y_	5%	176.04	9.27	7.42	1.85	5.56	3.71	9.27	264.70
JZ15B_x_C_y_	15%	157.51	27.80	22.24	5.56	16.68	11.12	27.80	264.70
JZ25B_x_C_y_	25%	138.98	46.33	37.06	9.27	27.80	18.53	46.33	264.70
JZF5	5%	185.30	-	-	-	7.94	5.29	-	251.47
JZF15	15%	185.30	-	-	-	23.82	15.88	-	225.00
JZF35	35%	185.30	-	-	-	55.59	37.06	-	172.06
JZF100	100%	185.30	-	-	-	158.82	105.88	-	0

**Table 2 materials-16-04233-t002:** Chemical composition and content of RBP and RCP.

Material Type	Chemical Composition and Content (wt%)							
	CaO	SiO_2_	MgO	Al_2_O_3_	Fe_2_O_3_	K_2_O	Na_2_O	SO_3_
Cement	52.30	21.09	1.20	5.00	3.89	0.68	0.39	2.70
Fly ash	4.15	62.4	0.55	23.7	6.89	0.97	1.23	/
RBP	3.11	60.92	1.98	17.44	5.58	3.14	2.40	0.081
RCP	21.74	47.86	2.67	10.07	4.41	1.38	0.489	1.29

**Table 3 materials-16-04233-t003:** X-ray diffraction peaks of Ca(OH)_2_ and SiO_2_.

Type	CH (18.1°)	CH (34.0°)	SiO_2_ (26.7°)	*R* Value	Compressive Strength (MPa)
CP	288	240	188	1.622	15.12
CFBP	522	330	426	2.138	9.64
CFCP	460	314	482	1.979	10.97
CFBCP	428	303	362	1.909	11.56
CFP	507	260	697	2.635	7.36

## Data Availability

The data used to support the findings of this study are available from the corresponding author upon request.

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
