# Peer review of "Experimental Study on the Strength and Hydration Products of Cement Mortar with Hybrid Recycled Powders Based Industrial-Construction Residue Cement Stabilization of Crushed Aggregate"

_materials, 2023, doi:10.3390/ma16124233_

Round 1

Reviewer 1 Report

The manuscript entitled “Experiment study on the strength properties and hydration products of cement mortar with hybrid recycled powders” presented experimental study. The influence of different parameters was studied and analyzed. However, there seems to be little to no novelty in this study. This reviewer recommends major editing and resubmits for re-review.

Comments:

• The English writing of the manuscript needs improvement. Therefore, it could benefit greatly from professional editing to improve technical writing and English.

• Please mention your study limits and suggest some future research topics

• In References, the sources are written in different styles. Please update the reference list. It is necessary to bring in accordance with the requirements of the magazine for the design of References. If possible, indicate DOI.

• Please use some innovative keywords.

• Please mention your study limits in the abstract.

• The literature can be expanded by studying some of these papers. 

• Effect of recycled coarse aggregate and bagasse ash on two-stage concrete

• Life cycle impact assessment of recycled aggregate concrete, geopolymer concrete, and recycled aggregate-based geopolymer concrete

• The Conclusions should reflect what the practical application of the results obtained in this study is. In what climatic conditions should the recommendations of the authors be taken into account?

• The authors should increase their discussion on previous related research and highlight how their study is providing a different approach or adding significantly to what has been done. The authors have to explain what is the new here in comparison with the previous studies. The novelty of the current work should be highlighted in the introduction. Please try to mention a problem that needs solving - in other words, the research question underlying your study clearer.

• The title of the manuscript should be revised.

• Some types of standards should be used to perform different experimental studies. Please provide details for the standards used in each study.

• Section 4 should be discussed in detail.

• The authors must redo the Abstract and bring it in compliance with the requirements of the journal. The scientific problem is poorly described (Background). The scientific novelty is not indicated. I recommend shortening the Abstract to 200 words. Editors strongly encourage authors to use the following style of structured abstracts, but without headings: (1) Background: Place the question addressed in a broad context and highlight the purpose of the study; (2) Methods: Briefly describe the main methods or treatments applied; (3) Results: Summarize the article's main findings; and (4) Conclusions: Indicate the main conclusions or interpretations. The abstract should be an objective representation of the article

• It is advisable to add a flowchart at the beginning of the paper. Then the article would become more visual and structured

• The economic aspects are also required for sustainability in social aspect. It is suggested to authors to evaluate the cost-benefit study of this as a further investigation

• The conclusion should be an objective summary of the most important findings in response to the specific research question or hypothesis. A good conclusion states the principal topic, key arguments and counterpoint, and might suggest future research. It is important to understand the methodological robustness of your study design and report your findings accordingly. Please improve your conclusion section.

• The English writing of the manuscript needs improvement. Therefore, it could benefit greatly from professional editing to improve technical writing and English. 

Author Response

Thank you for considering our manuscript (Materials-2389209). Your comments has been very helpful for improving the quality of our paper and thus have been highly appreciated. Changes have been made to the manuscript according to your comments. The comments have been addressed and presented below.

Point 1.The English writing of the manuscript needs improvement. Therefore, it could benefit greatly from professional editing to improve technical writing and English.

Response 1:Thanks for your suggestion. We feel really sorry for our poor writings, however, we have checked the grammar carefully of the overall paper. We employed an English-language editing service, American Journal Experts(AJE), to polish article. Due to AJE’s help, the article was edited extensively. And we hope the revised manuscript could be acceptable for you.Certification is attached.

Point 2.Please mention your study limits and suggest some future research topics.

Response 2:Thanks for your advice.The semi-rigid base is used as a structural bearing stratum in the road with the inorganic binder, compared with the flexible road that adopts the granular material as road construction layer, having the advantage of high strength, carrying capacity, good wholeness, and rigid, accounting for more than 85% of the road base engineering in China [1]. Fly ash as a by-product of coal-fired power stations causes significant economic and environmental problems. However, adding some fly-ash into cement stabilization of crushed aggregate(CSCA) can effectively promote hydration,less crack and improve interface adhesion and later strength to realize the first leap of turning industrial solid wastes into treasure[2].Brick-concrete composite micro-powder (RBCP) is used to replace part of cement which not only makes effective use of concrete and brick waste(C&BW) but also meets the requirements of green development strategy. Therefore,this paper did not evaluate the shrinkage performance of cement stabilized mortar containing fly ash and RBCP.

Based on the physical and chemical characteristics of low-quality fly ash and RBCP, this paper will use chemical compound excitation method to stimulate the activity of low-quality fly ash and RBCP, which can greatly increase the content of fly ash and RBCP in cement. Through the optimization design of material composition and the study of cooperative development law, the pavement base cement with retarding micro-expansion characteristics will be prepared to alleviate the water-stabilized foundation. The cracking of the layer and the pavement performance of the special cement for the base of this pavement will be studied.

  • X.W. Li,M.M. “Experimental research on pavement performance of cement stabilized base recycled mixture,”Applied Mechanics and Materials,vol.94-96,Article ID: 130.88.90.140,2011.
  • X. Chen,M.K. Zhou,J.Liuet al.“Decoupling analysis of fly ash effects in cemnet-fly ash stabilized crushed stones,”Journal of Building Materials,6,no.13,p

p.764-768,2010.

Point 3.In References, the sources are written in different styles. Please update the reference list. It is necessary to bring in accordance with the requirements of the magazine for the design of References. If possible, indicate DOI.

Response 3:Thanks for point out this.We have reorganized this section according to the requirements of the magazine,which is highlighted in red in lines 536-674.

Point 4.Please use some innovative keywords.

Response 4:Thank you for your reminding. These some innovative keywords have been additional in revised manuscript, which is highlighted in red in lines 29-30.

Key words:Cement fly ash mortar ; hybrid recycled powder(HRP);mix ratio dedign;mechanical properties; hydration product;

Point 5.The literature can be expanded by studying some of these papers. 

  • Effect of recycled coarse aggregate and bagasse ash on two-stage concrete
  • Life cycle impact assessment of recycled aggregate concrete, geopolymer concrete, and recycled aggregate-based geopolymer concrete.

Response 5:Thanks for your valuable suggestion. The authors carefully read your recommended literature. The comparison is added in revised manuscript and highlighted in red.

In addition, with the help of an alkaline activator, it has become a research hotspot in the past decade to completely replace Portland cement with such auxiliary cementitious materials (SCMS), which has led to the development of geopolymer concrete [6, 7, 8].

  1. Lahiba Imtiaz,Sardar Kashif-ur-RehmanWesam,Salah Alaloul,et,al.Life cycle impact assessment of recycled aggregate concrete, geopolymer concrete, and recycled aggregate-based geopolymer concrete.Sustainability,2021, 13, 13515. https://doi.org/10.3390/su132413515.

Mosavi et al conducted a comparative study on the performance of recycled concrete formed by the pressure drop method and recycled concrete formed by the mechanical method according to the above theory. The research results showed that the pressure drop forming method can effectively improve the mechanical properties of recycled concrete.

9.Muhammad Faisal Javed,Afaq Ahmad Durrani,and Amir Mosavi,et al.Effect of recycled coarse aggregate and bagasse ash on two-stage concrete.Crystals, 2021, 11, 556. https://doi.org/10.3390/cryst11050556

Point 6.The authors should increase their discussion on previous related research and highlight how their study is providing a different approach or adding significantly to what has been done. The authors have to explain what is the new here in comparison with the previous studies. The novelty of the current work should be highlighted in the introduction. Please try to mention a problem that needs solving - in other words, the research question underlying your study clearer.

Response 6:We are very grateful for your professional review work on our article.The

introduction has been rewritten in revised manuscript, which is highlighted in red.

Since 2014, the total amount of construction and demolition waste (C&DW)produced in China has been estimated to be about 1.55 ~ 2.4 billion tons. Between concrete and brick waste possesses one of the highest shares and accounts for about 85% of the total of C&DW. Seeking "recycling of (C&DW)" has become the most effective environmental protection method. The semi-rigid base is used as a structural bearing stratum in the road with the inorganic binder, compared with the flexible road that adopts the granular material as road construction layer, having the advantage of high strength, carrying capacity, good wholeness, and rigid, accounting for more than 85% of the road base engineering in China . Fly ash as a by-product of coal-fired power stations causes significant economic and environmental problems. However, adding some fly-ash into CSCA can effectively promote hydration and improve interface adhesion and later strength to realize the first leap of turning industrial solid wastes into treasure. Furthermore, the addition of recycled brick-concrete composite micro-powder (RBCP) in the CSCA is used to replace part of cement, which not only makes effective use of concrete and brick waste(C&BW) but also meets the requirements of green development strategy. Take a second-class highway with a thickness of 20cm for road base as an example, 1728m3/km recycled aggregate can be used for cement stabilized materials(CSCA).

Point 7. The title of the manuscript should be revised.

Response 7:Firstly, the authors would like to thank you for the valuable comments on the manuscript. Thanks for point this problem out,the author has carefully read the manuscript and revised the improper expression of the article. The revised title is revised “Experimental study on the strength and hydration products of cement mortar with hybrid recycled powders based industrial residue cement stabilization of crushed aggregate (IRCSCA)”.

Point 8.Some types of standards should be used to perform different experimental studies. Please provide details for the standards used in each study.

Response 8:Thank you for the valuable suggestion on the part of 2.4.These following information has been added in revised manuscript, which is highlighted in red in the revise manuscript.

The flexural strength and unconfined compressive strength of the analyzed cement mortar samples were measured according to GB/T17671-2021[37], respectively. Flexural strength and uniaxial compressive loads were applied at a loading rate of 50N/s and 2500N/s separately until the specimens failed.

Point 9.The authors must redo the Abstract and bring it in compliance with the requirements of the journal. The scientific problem is poorly described (Background). The scientific novelty is not indicated. I recommend shortening the Abstract to 200 words. Editors strongly encourage authors to use the following style of structured abstracts, but without headings: (1) Background: Place the question addressed in a broad context and highlight the purpose of the study; (2) Methods: Briefly describe the main methods or treatments applied; (3) Results: Summarize the article's main findings; and (4) Conclusions: Indicate the main conclusions or interpretations. The abstract should be an objective representation of the article.

Response 9:We are very grateful for your professional review work on our article. The abstract has been rewritten in revised manuscript, which is highlighted in red.

The strength formation mechanism for industrial-construction residue cement stabilization of crushed aggregate (IRCSCA) is is not clear. To expand the application range for recycled micropowders in road engineering,the dosages of eco-friendly hybrid recycled powders (HRPs) with different proportions of RBP and RCP affect the strengths of the cement-fly ash mortar at different ages,and the strength formation mechanism were studied with X-ray diffraction (XRD) and scanning electron microscopy (SEM).The results showed that the early strength of the mortar was 2.62 times higher than that of the reference specimen when a 3/2 mass ratio of brick powder and concrete powder was mixed to form the HRP and replace some of the cement. With increasing HRP content substituted for the fly ash, the strength of cement mortar first increased and then decreased. When the HRP content was 35%, the compressive strength of the mortar was 1.56 times higher than that of the reference specimen, and the flexural strength was 1.51 times higher; XRD and SEM studies of the hydrated cement mixed with HRP showed that the amount of CH in the cement paste was reduced by the pozzolanic reaction of HRP at later hydration ages,and it was very useful in improving the compactness of the mortar.The XRD spectrum of the cement paste made with HRP indicated that the CH crystal plane orientation index R, with a diffraction angle peak of approximately 34.0, was consistent with the cement slurry strength evolution law, and this research provides a reference for the application of HRP to produce IRCSCA.

Point 10.It is advisable to add a flowchart at the beginning of the paper. Then the article would become more visual and structured.

Response 10:Thanks for your suggestion.The graphical abstract will be added to the beginning of the article.

Point 11.The economic aspects are also required for sustainability in social aspect. It is suggested to authors to evaluate the cost-benefit study of this as a further investigation.

Response11:Thank you for your reminding. We added corresponding sustainability in real transportation infrastructure construction aspect.The following statements have been added to the revised manuscript and highlighted in red in lines 66-68.

Take a second-class highway with a thickness of 20cm for road base as an example, 1728m3/km recycled aggregate can be used for cement stabilized materials(CSCA). Therefore, it is of great theoretical and engineering significance to carry out the research on CSCA contained RBCP and RBCA.

Point 12.The conclusion should be an objective summary of the most important findings in response to the specific research question or hypothesis. A good conclusion states the principal topic, key arguments and counterpoint, and might suggest future research. It is important to understand the methodological robustness of your study design and report your findings accordingly. Please improve your conclusion section.The Conclusions should reflect what the practical application of the results obtained in this study is. In what climatic conditions should the recommendations of the authors be taken into account? Section 4 should be discussed in detail.

Response12:Thanks for your comments. The authors carefully read the conclusion part. The sentence has been rephrased, which was highlighted in red in the revised manuscript.

The utilization of construction solid waste is an effective aspect of green and sustainable economic development. The comprehensive experimental study described herein indicates the viability of green recycling and using HRP to replace Portland cement and fly ash. The mechanical and hydration mechanisms of the cement-fly ash mortar were investigated by varying the proportions of RBP, RCP in HRP and the replace ratio of HRP . Based on the findings of this study, the following conclusions are drawn.

(1) HRP has specific activities and can be used as auxiliary cementitious materials to replace some of the cement. In the presence of fly ash with increasing HRP contents, the strength of cement mortar first increased and then decreased, and the early strengths of the cement mortar specimens with single RBP were better than those of the cement mortar specimens with RCP. When the single content was lower than 15%, the maximum increase in the 3-d compressive strength was 19.08%, and the maximum decrease in the 28-d compressive strength was 2.39%.

(2) HRP can be used as mineral admixtures to replace some of the fly ash. They were mixed with a mass ratio of 3/2 to form the HRP, which effectively improved the strength of the mortar specimen. When the RBP, RCP, and fly ash were mixed with the mass ratio 21/14/65, the flexural strength of the mortar was 1.51 times higher than that of the cement fly ash specimen, and the compressive strength was 1.56 times higher at 28 d.

(3) In the presence of fly ash, the HRP mainly exerted the microaggregate effect and promoted the cement hydration reaction, which significantly improved the microstructural compactness of the mortar and the thickness of the C-S-H on the surfaces of the fly ash particles.

(4)The pozzolanic effect operating on the HRP in the later stages reduced the intensities of the CH diffraction peaks. The crystal plane orientation index R with a diffraction angle of 34.0° (101) was consistent with the evolution law for the compressive strength.

Reviewer 2 Report

The authors used waste powders as replacement of cement. The paper is generally good but it needs improvement. Followings should be carried out before acceptance:

The abstract should contain important results of the study.

Is the materials in Fig 1. are the materials used in the experiments?

EDX analyses also can be added

The following studies can be added to line 40 for references 2and3:  production of perlite-based-aerated geopolymer using hydrogen peroxide as eco-friendly material for energy-efficient buildings

Replacement of cement can be extended with glass, marble, coal bottam ash and etc. The followings can be added to line 42 in references 4-5:  influence of replacing cement with waste glass on mechanical properties of concrete; 

Novelty is not clear. Very same studies are already exists. What is the difference?

The reason for selecting design mixture should be added.

Comments on Table 2 should be improved.

Compare your results with existing studies

Add photos for test setup?

Quality of Fig 3. should be improved.

Comments on SEM analyses should be extended.

Please add damaged photos damaged photos of samples

Add some summary for conclucision

Add recent studies on this subject to introduction. There are many studies on the introduction for this topic.

Conclusion should be improved. The recommendation consdiering all test should be given for engineers.

Author Response

Point 1.The abstract should contain important results of the study.

Response 1:We are very grateful for your professional review work on our article. The abstract has been rewritten in revised manuscript, which is highlighted in red.

The strength formation mechanism for industrial-construction residue cement stabilization of crushed aggregate (IRCSCA) is is not clear. To expand the application range for recycled micropowders in road engineering,the dosages of eco-friendly hybrid recycled powders (HRPs) with different proportions of RBP and RCP affect the strengths of the cement-fly ash mortar at different ages,and the strength formation mechanism were studied with X-ray diffraction (XRD) and scanning electron microscopy (SEM).The results showed that the early strength of the mortar was 2.62 times higher than that of the reference specimen when a 3/2 mass ratio of brick powder and concrete powder was mixed to form the HRP and replace some of the cement. With increasing HRP content substituted for the fly ash, the strength of cement mortar first increased and then decreased. When the HRP content was 35%, the compressive strength of the mortar was 1.56 times higher than that of the reference specimen, and the flexural strength was 1.51 times higher; XRD and SEM studies of the hydrated cement mixed with HRP showed that the amount of CH in the cement paste was reduced by the pozzolanic reaction of HRP at later hydration ages,and it was very useful in improving the compactness of the mortar.The XRD spectrum of the cement paste made with HRP indicated that the CH crystal plane orientation index R, with a diffraction angle peak of approximately 34.0, was consistent with the cement slurry strength evolution law, and this research provides a reference for the application of HRP to produce IRCSCA.

Point 2.Is the materials in Fig 1. are the materials used in the experiments?

Response 2:In order to explain the necessity of this manuscript more effectively,the author reorganizes this part of the 2.1 .The following statements have been added to the revised manuscript and highlighted in red in page 4 line 161-178.

The C&BW used to prepare the RBP and RCP was collected after demolition of a 50 year-old factory district located in Er-qi town in Zhengzhou, Henan Province. This waste consisted of clay bricks, concrete, ceramics and so on. Ordinary Portland cement 42.5 was prepared according to the Chinese standard JTG/T F20-2015[50], and tap water (with a temperature of ~20℃) was used in the experiments. The sand used in the mortars was river sand with a modulus of 2.9.

The HRP were produced with a three-stage process after the original waste clay brick and concrete was processed. In the first stage, a crusher was used to reduce the sizes of waste bricks and concrete to fine the particles to below 10 mm, and particles with different particle sizes were prepared via passage through a vibrating screen; these were categorized as recycled coarse aggregate. In the second stage, the product from stage one was fed into a ball grinding mill with a circular cavity to produce fine aggregates with maximum sizes of 2.36 mm, which was categorized as recycled fine aggregate. The output of stage two was then fed into an electromagnetic sample pulverizer to produce fine powders with maximum sizes of 45 μm, which were categorized as HRP. Fig. 1 describes the method used to prepare the HRP through deep processing of the C&BW. The total processing time, including cruising, sieving and grinding, was approximately 6–8 minutes.

Point 3.EDX analyses also can be added.

Response 3.We appreciate your suggestions. We agree with you very much.At that time, these samples could not be completed for testing due to the outbreak of the COVID-19(XBB.1.5) epidemic. After receiving your suggestion, we actively prepared for the test, but the test sample could not be saved in time to complete the test.We have scheduled to do this test.The results of the test will be returned around May 20, as the lab needs to be booked.However, the editorial board has asked for a revised manuscript to be returned on May 15, and we are waiting for the article to be accepted to add the results of that section.

Point 4. Replacement of cement can be extended with glass, marble, coal bottam ash and etc. The followings can be added to line 42 in references 4-5: influence of replacing cement with waste glass on mechanical properties of concrete; 

Response 4.Thank you very much for your suggestion Already in lines 4-5 of references can increase the impact of waste glass.

Incorporating recycled glass into concrete enhanced rheological properties, thereby decreasing the need for chemical admixtures and increasing cost-effectiveness

Point 5.Novelty is not clear. Very same studies are already exists. What is the difference?Compare your results with existing studies.

Response 5.Therefore, the pozzolanic characteristics of the RBP and RCP show that the HRP can be used as a cement supplement. Based on this, more and more interest has been attracted to exploring the use of the HRP. In addition, it can be seen that fly ash can be used as a concrete admixture and an auxiliary cementing material in the cement stabilization of crushed aggregate (CSCA) in the road engineering. However, there are few reports on the interaction between solid waste materials and cement. Moreover, there are few reports on the influence of RBP, RCP, and fly ash on the strength of cement mortar and the hydration mechanism. To respond to this need, in the presence of fly ash, it is of theoretical significance to explore the comprehensive effect of RBP and RCP and its in-fluence on the strength formation mechanism of cement mortar.

Point 6.The reason for selecting design mixture should be added.

Response 6.Thank you very much, the reason for designing the mixture has been updated on line 180.

The RBP and RCP composition is similar to that of fly ash, and the high contents of SiO2 and Al2O3 in RBP and RCP help promote pozzolanic activity ; therefore, it is expected that the RBP and RCP with an appropriate particle size can be used as an SCM in concrete. RBP and RCP has a relatively high reactivity after deep grinding; thus, RBP and RCP is utilized as an SCM in concrete preparation. In addition, the micro-performance, mechanical properties and durability of concrete containing RBP and RCP have been investigated by scholars worldwide.

Point 7.Comments on Table 2 should be improved

Response 7.Thank you very much for your professional review of our articles. The description of Table 2 has been rewritten in the revised version, highlighted in red

Out display.

Table 2 shows that the SiO2 content of the RBP was close to that of fly ash, and the contents of CaO, Al2O3 and Fe2O3 were between those of the cement and fly ash, which implied that RBP had a good oxide distribution. Based on the diffraction peaks shown in Fig. 4, RBP contained SiO2 crystals, which can be combined with the Ca(OH)2 generated by cement hydration to generate C-S-H and C-A-S-H, calcium sulfoaluminate hydrate[33,54,55]. Li et al. also indicated that fine SiO2 increased the compactness and strength of a mortar sample. Table 2 shows that the RPC contained CaO, which reacts with water to form the alkali activator Ca(OH)2, and Ca(OH)2 improves the activity of the RBP. Moreover, both the RBP and RCP contained SiO2 and Al2O3. Hence, both powdered materials exhibited pozzolanic activity, which effectively improved the late strengths of the mortars. In addition, there was a small amount of albite in the recycled concrete powder, which may have arisen from mixing of fine aggregate stone chips; additionally, the recycled brick powder contained a large amount of albite, and albite was one of the main components of the clay bricks. The CaCO3 in RCP can shorten the induction period of C3S and participate in the hydration reaction to form carboaluminates, which refines the pore structures of cement pastes [56]. However, the diffraction peaks for hydrated calcium silicate and hydrated calcium aluminate were not found in the XRD pattern for RCP, indicating that the cement particles in RCP were hydrated. In summary, RBP and RCP showed specific activities and replaced some of the cementitious materials or mineral admixtures.

Point 8.Add photos for test setup?Quality of Fig 3. should be improved.

Response 8. Thank you very much.The clarity of adding photos (Figure 3) to the test setup has been improved .

Point 9.Comments on SEM analyses should be extended.Please add damaged photos damaged photos of samples.

Response 9. Thank you for the valuable suggestion on the part of 3.5.1.These following information has been added in revised manuscript, which is highlighted in red in the revise manuscript. which is highlighted in red in lines 399-421. We feel really sorry for our careless in experiment, and we cannot add photos of test sample damage to the article.

Fig. 8(a) shows that white flocculent substances were attached to the surfaces of the fly ash spherical particles in the cement-fly ash mortar. This material is a hydraulic cementitious material, such as C-S-H and C-A-H, and it is formed by reactions between SiO2 and Al2O3 in the fly ash with the cement hydration product Ca(OH)2, which is consistent with the conclusions of Du[58], Hardjito[59] and Narmluk[60]. Needle-like substances appeared between the spherical particles of the fly ash, and these comprised ettringite (AFt) formed by the reaction of Al2O3 and Ca(OH)2 in fly ash. A comparison of Fig. 8 (a) and Fig. 8(b) showed that there were holes in the internal micromorphologies of the cement-fly ash mortar made with HRP and the cement-fly ash mortar made without HRP. However, the cement-fly ash mortar containing HRP was relatively forming a stable dense structure, which showed that HRP filled the microaggregates. In addition, the CaO in the RCP reacted with water to generate the alkali activator Ca(OH)2, which promoted a secondary hydration reaction between the active component of the RBP and the cement hydration product. The combined effects of RBP and RCP increased the hydration products formed in the mortar with HRP, and the thicknesses of white flocculated substances attached around the fly ash particles in Fig. 8(b) were larger than those in Fig. 8(a). Furthermore, many fibrous crystals were interspersed between the hydration products, such as C-S-H gel in Fig. 8(b), to improve the compactness of the cement paste. The combined effects of HRP-Ⅰ improved the strength of the cement-fly ash mortar. Fig. 8(c)[Element map] shows the elemental map of cement-fly ash mortar and cement-fly ash mortar with HRP; the calcium content decreases and the silicon content increases as the HRP replacement ratio increases because HRP contains lower calcium and higher silicon contents than cement.

Point 10.Add some summary for conclucision.Conclusion should be improved. The recommendation consdiering all test should be given for engineers.

Response 10.Thanks for your comments. The authors carefully read the conclusion part. The sentence has been rephrased, which was highlighted in red in the revised manuscript.

The utilization of construction solid waste is an effective aspect of green and sustainable economic development. The comprehensive experimental study described herein indicates the viability of green recycling and using HRP to replace Portland cement and fly ash. The mechanical and hydration mechanisms of the cement-fly ash mortar were investigated by varying the proportions of RBP, RCP in HRP and the replace ratio of HRP . Based on the findings of this study, the following conclusions are drawn.

(1) HRP has specific activities and can be used as auxiliary cementitious materials to replace some of the cement. In the presence of fly ash with increasing HRP contents, the strength of cement mortar first increased and then decreased, and the early strengths of the cement mortar specimens with single RBP were better than those of the cement mortar specimens with RCP. When the single content was lower than 15%, the maximum increase in the 3-d compressive strength was 19.08%, and the maximum decrease in the 28-d compressive strength was 2.39%.

(2) HRP can be used as mineral admixtures to replace some of the fly ash. They were mixed with a mass ratio of 3/2 to form the HRP, which effectively improved the strength of the mortar specimen. When the RBP, RCP, and fly ash were mixed with the mass ratio 21/14/65, the flexural strength of the mortar was 1.51 times higher than that of the cement fly ash specimen, and the compressive strength was 1.56 times higher at 28 d.

(3) In the presence of fly ash, the HRP mainly exerted the microaggregate effect and promoted the cement hydration reaction, which significantly improved the microstructural compactness of the mortar and the thickness of the C-S-H on the surfaces of the fly ash particles.

(4)The pozzolanic effect operating on the HRP in the later stages reduced the intensities of the CH diffraction peaks. The crystal plane orientation index R with a diffraction angle of 34.0° (101) was consistent with the evolution law for the compressive strength.

Reviewer 3 Report

The article tries to explore the comprehensive effect of RBP and RCP and its influence on the strength formation mechanism of cement mortar. The experimental activity was well-conceived and performed correctly. Nevertheless, some issues are not correctly addressed, and the article needs revisions before being suitable for publication. Some detailed comments are provided below:

·       Abstract: All the acronyms have to be defined the first time you use them (RBP, RCP, CH).

·       Lines 25-26: This sentence is not connected with the rest of the abstract and it makes no sense. Reduction of CH content does not improve microstructure.

·       Lines 35-36: Cement production accounts for 8% of total CO2 emissions. The authors should find updated references for that.

·       Lines 91-93: CH does not stand for cement hydration products. Please change that.

·       Lines 180: Why did the authors choose those percentages? Skipping from 35% to 100% with no intermediate percentage seems weird.

·       Figure 2: You should label the mixtures in the Figure.

·       Lines 208-209: Testing ages must be specified for all different tests.

·       Lines 225: A particle size distribution cannot be “better” than another one without explanation.

·       Figures 5 and 7: It is so difficult to see different testing ages results… Maybe different colors or other types of shading (not only stripes) will help.

·       Lines 287: What is the reason for conducting flexural strength?

·       Lines 380: This title is a bit strange since you already spoke about mixture HRP-I before, so maybe can be changed to “further microstructural analysis” or so.

·       Overall: Please use an English native speaker to check the manuscript. There are many inconsistencies and/or grammatical errors throughout the manuscript.

Overall: Please use an English native speaker to check the manuscript. There are many inconsistencies and/or grammatical errors throughout the manuscript.

Author Response

  Point 1. Abstract: All the acronyms have to be defined the first time you use them (RBP, RCP, CH).Lines 25-26: This sentence is not connected with the rest of the abstract and it makes no sense. Reduction of CH content does not improve microstructure.

Response 1.We are very grateful for your professional review work on our article.In order to explain the necessity of this manuscript more effectively,the author reorganizes this part of the essay , “In this paper, based on the strength of cement-fly ash mortar, the dosage of the eco-friendly hy-brid recycled powder (HRP) with the different proportions of recycled brick powder (RBP )and recycled concrete powder (RCP) affects the compres-sive and flexural strength of cement-fly ash mortar at different ages.” “The XRD spectrum of the cement paste with the HRP indicated that the calcium hydroxide (CH) crystal plane orienta-tion index R with a diffraction angle peak around 34.0 is consistent with the cement slurry strength evolution law, and this research provided the reference for application of HRP to pro-duce IRCSCA.”And we deleted“Lines 25-26” and this sentence has been changed in revised manuscript.  

Point 2. Lines 35-36: Cement production accounts for 8% of total CO2 emissions. The authors should find updated references for that.Lines 91-93: CH does not stand for cement hydration products. Please change that.

Response 2.Firstly, the authors would like to thank you for the valuable comments on the manuscript. The sentence has been rephrased, and the following statements have been added to the revised manuscript and highlighted in red in page 1 line 35-36 and page 2 line 91-93.

“In cement production, the greenhouse gases produced to account for 8% of global greenhouse gas emissions. [1]”

  • Sousa,V.; Bogas, J. A.; Journal of Cleaner Production: Comparison of energy consumption and carbon emissions from clinker and recycled cement production. 2021, 306, 127277.

Point 3.Lines 180: Why did the authors choose those percentages? Skipping from 35% to 100% with no intermediate percentage seems weird. Figure 2: You should label the mixtures in the Figure.

Response 3.Thanks for point out this.The modifications have been corrected and highlighted in red revised manuscript.  

Under the same alternative mass fraction, with the increase of curing age, the strength of recycled composite micronized cement sand gradually increased in the presence of fly ash, and when the regenerated composite powder completely replaced fly ash, the strength of the rubber sand specimen 3 and 7d increased and the strength decreased by 28d, which indicated that the regenerated composite micropowder mainly played a microaggregate filling effect in the early strength formation of the specimen, and with the gradual extension of the age, the later strength decrease was not significant because the active ingredient could react with cement hydration products. The plastic properties of rubber sand are improved.

Point 4.Lines 208-209: Testing ages must be specified for all different tests.Lines 225: A particle size distribution cannot be “better” than another one without explanation.

Response 4.In order to further analyze the influence mechanism of HRP on the strength formation of cement fly ash mortar, combined with Table 1, cement fly ash slurry (CP) was used as the benchmark, cement fly ash recycled brick slurry (CFBP), cement fly ash recycled concrete slurry (CFCP) and cement fly ash recycled brick powder concrete slurry (CFBCP) as the research objects. Paste specimens were prepared with the NJ-160B mixing mechanism and modeled in rectangular molds measuring 40 mm × 40 mm × 160 mm. The curing conditions of the paste are similar to the curing conditions of mortar, and the relative compressive strength of composite micronized concrete with different substitution rates is 3、7 and 28d.

The particles of recycled brick powder are finer and evenly distributed, the particles of recycled concrete powder are coarser, the particle size of the former is smaller than the latter, and the particle size distribution of RBP is better than that of RCP.

Point 5.Figures 5 and 7: It is so difficult to see different testing ages results… Maybe different colors or other types of shading (not only stripes) will help.Lines 287: What is the reason for conducting flexural strength?Lines 380: This title is a bit strange since you already spoke about mixture HRP-I before, so maybe can be changed to“further microstructural analysis”or so.

Response 5.Thank you for your reminding. We reorganized Figure 2.The following statements have been added to the revised manuscript and highlighted in red.

In road projects, cement fly ash stabilized gravel base is mainly subjected to bending and tensile stresses under the action of vehicle loads.In this paper, the prismatic specimens were formed in the laboratory and the flexural and tensile strength test of the cement sand was carried out by the flexural testing machine, so it was written as flexural strength in the article.

Point 6. Overall: Please use an English native speaker to check the manuscript. There are many inconsistencies and/or grammatical errors throughout the manuscript.

Response 6. Thanks for your suggestion. We feel really sorry for our poor writings, however, we have checked the grammar carefully of the overall paper. We employed an English-language editing service, American Journal Experts(AJE), to polish article. Due to AJE’s help, the article was edited extensively. And we hope the revised manuscript could be acceptable for you.Certification is attached.

Round 2

Reviewer 1 Report

Accept

Author Response

Thank you for considering our manuscript (Materials-2389209). All author would like to take this opportunity to express our sincere gratitude for your time and efforts in reviewing our article. Your positive feedback and encouragement have been instrumental in enhancing the quality of the final manuscript, and we are truly honored to have had the benefit of your expertise and insights.

Reviewer 2 Report

The authors improved very well paper.

But there are still some issues befor publication

Many mistakes are avaliable in the text: See error!Reference source not found. There is 43 times for this error. No need for (IRCSCA) in title. It is not adequate

Is it possible to increase the size of photos in Figure 1. Can you increase the quality of Figure 2. Do not reduce the size of Figure 2.

Check prevous question point 4. See the recommended one. In addition to this one, use of waste glass powder toward more sustainable geopolymer concrete; flexural behavior of reinforced concrete beams using waste marble powder towards application of sustainable concrete can be added.

Size of Figure 3 should be increased. Quality of Figures 6 and 7 should be added. It is hard to read the writings on Fig. 6 and 7. Graphs on Fig8 is not readeable. Please increase the size or punto of writing.

Author Response

Thank you for considering our manuscript (Materials-2389209). Your comments has been very helpful for improving the quality of our paper and thus have been highly appreciated. Changes have been made to the manuscript according to your comments. The comments have been addressed and presented below.

  • Many mistakes are avaliable in the text: See error!Reference source not found. There is 43 times for this error. No need for (IRCSCA) in title. It is not adequate
  • We really appreciate your professional review work on our articles. In accordance with your comments, we have revised the title.
  • Is it possible to increase the size of photos in Figure 1. Can you increase the quality of Figure 2. Do not reduce the size of Figure 2.
  • We really appreciate your professional review work on our articles. In accordance with your comments, we have revised the title.
  • Thank you for your valuable comments, we have adjusted Figure 2.
  • Check prevous question point 4. See the recommended one. In addition to this one, use of waste glass powder towar d more sustainable geopolymer concrete; flexural behavior of reinforced concrete beams using waste marble powder towards application of sustainable concrete can be added.

We really appreciate your professional review work on our articles. In order to explain more effectively the necessity of this manuscript, the author has made a new addition to this part of the article.

[7]. "Flexural behavior of reinforced concrete beams using waste marble powder towards application of sustainable concrete" by Memduh Karalar, Yasin Onuralp Özkılıç , Structural Materials

Volume 9 - 2022 | https://doi.org/10.3389/fmats.2022.1068791

  • The addition of waste marble powder can significantly improve the bending performance of concrete beams, while also helping to reduce environmental pollution and resource waste.
  • Size of Figure 3 should be increased. Quality of Figures 6 and 7 should be added. It is hard to read the writings on Fig. 6 and 7. Graphs on Fig8 is not readeable. Please increase the size or punto of writing.
  • Thanks for the reminder. We resized Figure 3, the quality of Figure 6 and Figure 7, and the size and font of Figure 8.

Reviewer 3 Report

After revision, the authors have addressed some of the comments, improving the article. However, some issues are still present:

·       There are many “Error! Reference source not found.”. Please modify that.

·       Skipping from 35% to 100% with no intermediate percentage seems weird. Can the authors explain this?

·       A particle size distribution cannot be “better” than another one without explanation. Can the authors explain this?

Author Response

Thank you for considering our manuscript (Materials-2389209). Your comments has been very helpful for improving the quality of our paper and thus have been highly appreciated. Changes have been made to the manuscript according to your comments. The comments have been addressed and presented below.

Point1.There are many “Error! Reference source not found.Please modify that.

Response 1.Thanks for point out this.We have reorganized this section according to the your suggestion.

Point2.Skipping from 35% to 100% with no intermediate percentage seems weird. Can the authors explain this?

Response 2.Thanks for point this problem out, the reason should be explained by the following.

The cement-fly ash mortar sample without recycled materials is named as reference specimen, which denoted as JZ. However, the particle size of HRP is more significant than that of cement and fly ash. Furthermore, the standard Chinese GB1596(2015) specifies a minimum cumulative content of 88% for particle-sized less than 45μm in the Class I powders. Compared with fly ash and cement, the application of HRP produced by current technology is limited. The main reason is that its performance is poor, and its particle size is large. Therefore, many researchers began to study the preparation of more delicate recycled powder[1,4-5]. In recent years, with the increasing engineering demand, the method of developing auxiliary cementitious materials has been widely concerned. Standard methods include the thermal process, physical form, chemical method, etc[2,5]. Due to its fineness and thus potential reactivity, a lot of pure and application-oriented fundamental research has been performed to scientifically and technologically support the HRP to replace partially Portland cement about road base in recent years[1-3]. In an investigation by Xiao et al. [6], after recycled concrete micro-powder replaces part of Portland cement, it could promote hydration reaction. The mechanical property test shows that the optimal proportion of recycled micro-powder instead of Portland cement is 15% ~ 30%. Similar to recycled concrete micro-powder, the recycled brick powder is also used as a supplement in road bases because of its pozzolanic effect[5].In recent research[7], different types of clay bricks from European countries were collected and then ground into powder to replace the cement. It is found that recycled brick powder can effectively improve cement paste’s sulfate resistance, refine cement paste’s structure, reduce permeability and improve the characteristics of calcium silicate gel [8-10]. Therefore, an environment-friendly way is to reuse HRP likely fly ash to replace cement-based materials and be widely used in road engineering.Therefore, the authors directly mixed the composite micronized fly ash replacement from 35% to 100% in the course of the experiment.

[1]T. Meng, Y.P.Hong, K.J Ying et al.“Comparison of technical properties of cement pastes with different activated recycled powder from construction and demolition waste,”Cement and Concrete Composites,vol.120,Article ID:104065,2021.

[2]R. Serpell, M. Lopez.“Properties of mortars produced with reactived cementitious materials,”Cement and Concete Composites,vol.64,pp.16-26,2015.

[3]Q.Y. Li and H.Z.Quan,“Performance and application technology of recycled concrete,”China Building Industry Press,Beijing,2010. 

[4]Q.Tian,M.J. Qu ,M. Zhang,et al.“Research progress on activation way of recycled powder of waste concrete,Bulletin of the Chinese Ceramic Society,vol.39,no.8,pp.2476-2485,2020.

[5]Q.Tian, Z.M.Ma, H.X.Wu,et al.“The utilization of eco-friendly recycled powder from concrete and brick waste in new concrete:A critical review,Cement and Concrete Composites,vol.114,Article ID103807,2020,

[6]J.Z. Xiao, Z.M. Ma,T.B. Sui,et al.“Mechanical properties of conc-rete mixed with recycled powder produced from construction and demolition waste,”Journal of Cleaner Production,vol.188,pp.720-731,2018.

[7]S. Wild, A.Gailius,H.Hansen, et al.“Pozzolanicity properties of a variety of  European clay bricks,”Build Resarch&Information,vol25,no.3,pp.170-175,2010.

[8]L.Turanli,F.Bektas and P.J.Monteiro,“Use of ground clay brick as a pozzolanic material to reduce the alkali-silica reaction,”Cement and Concrete Composites,vol.33,pp.1539-1542,2003. 

[9]F. Afshinnia and A.Poursace,“The potential of ground clay brick to mitigate Alk-ali-Sillica reaction in mortar prepared with highly reactive aggregate,”Construction and Building Materials,vol.95,pp.164-170,2015.

[10]M.O.Farrell,S. Wild and B.B.Sabir,“Pore size distribution and compressive strength of waste clay brick mortar,”Cement and Concrete Composites,vol.23,pp.81-91,2001.

 Point3.A particle size distribution cannot be “better” than another one without explanation. Can the authors explain this?

Response 3.Thanks for point this problem out, the reason should be explained by the following.

A comparison of particle size distributions between cement and HRP is shown in Fig.3b. As regards Fig.3(b), the volume of HRP particles increased most significantly when the particle size was smaller than 10 mm, showing that there is a large volume of particles in this range. RBP and RCP also had a rapid increase than cement when the particle size was smaller than 10 mm. Besides, RCP and fly ash had a large volume of particles higher than 100 mm, illustrating the coarser particles of them. Fig. 3(b) gives also the relationship between the cumulative volume of the examined powders and the corresponding particle diameter. As compared with cement particles, the particle size of fly ash was significantly coarser due to the lack of deep processing. The cumulative particle size distributions of RCP were similar to that of cement, and the particle size of RBP was even much finer than that of cement, with the maximum particle size no large than 100 mm.

 The median diameters of different HRP and cement.It can be seen that RBP had the smallest mean diameter of 12.639μm, followed by RCP, which means that about half of the particles of RBP is lower than cement. The finer particle size will increase the specific surface area and the number of atoms on the surface, which will lead to an increase of the surface energy of the particles and improve the reactivity. The median diameters of RCP and fly ash were higher than cement. Fly ash had the largest mean diameter, which was 1.6 times of that of cement. RBP had smaller median diameters than RCP, which is mainly attributed to the strength and structure of the original sources of different HRP. Furthermore,the cumulative percentage of particles with sizes under 10μm was 41.49% for RBP compared with 32.52% for RCP.The volume-averaged particle size decreased from 23.828μm for RBP to 26.892 μm for RCP, and to 48.019μm for fly ash.Generally, the RBP is better than the RCP from the particle size distributions.

Round 3

Reviewer 3 Report

The authors correctly addressed most of my comments. I would recommend this article for publication.